# Emerging Nanomedicine Approaches in Targeted Lung Cancer Treatment

**DOI:** 10.3390/ijms252011235

**Published:** 2024-10-19

**Authors:** Isaic Alexandru, Lavinia Davidescu, Alexandru Cătălin Motofelea, Tudor Ciocarlie, Nadica Motofelea, Dan Costachescu, Monica Steluta Marc, Noemi Suppini, Alina Simona Șovrea, Răzvan-Lucian Coșeriu, Daniela-Andreea Bondor, Laura-Gabriela Bobeică, Andreea Crintea

**Affiliations:** 1Department X of General Surgery, “Victor Babes” University of Medicine and Pharmacy, 300041 Timisoara, Romania; isaic.alexandru@umft.ro; 2Department of Medical Disciplines, Faculty of Medicine and Pharmacy, University of Oradea, 410073 Oradea, Romania; 3Department of Internal Medicine, Faculty of Medicine, “Victor Babes” University of Medicine and Pharmacy, 300041 Timisoara, Romania; alexandru.motofelea@umft.ro; 4Department VII Internal Medicine II, Discipline of Cardiology, “Victor Babes” University of Medicine and Pharmacy, 300041 Timisoara, Romania; ciocarlie.tudor@umft.ro; 5Department of Obstetrics and Gynecology, “Victor Babes” University of Medicine and Pharmacy, Eftimie Murgu Sq. No. 2, 300041 Timisoara, Romania; nadica.motofelea@umft.ro; 6Radiology Department, “Victor Babes” University of Medicine and Pharmacy, 300041 Timisoara, Romania; costachescu.dan@umft.ro; 7Discipline of Pulmonology, “Victor Babes” University of Medicine and Pharmacy Timisoara, 300041 Timisoara, Romania; marc.monica@umft.ro (M.S.M.); noemi.suppini@umft.ro (N.S.); 8Department of Morphological Sciences, “Iuliu Hațieganu” University of Medicine and Pharmacy, 400349 Cluj-Napoca, Romania; simona.sovrea@umfcluj.ro; 9Department of Microbiology, University of Medicine, Pharmacy, Science and Technology “George Emil Palade”, 540142 Târgu-Mures, Romania; lucian-razvan.coseriu@umfst.ro; 10Department of Medical Biochemistry, “Iuliu Hațieganu” University of Medicine and Pharmacy, 400012 Cluj-Napoca, Romania; bondor.daniela.andre@elearn.umfcluj.ro (D.-A.B.); bobeica.laura.gabr@elearn.umfcluj.ro (L.-G.B.); crintea.andreea@elearn.umfcluj.ro (A.C.)

**Keywords:** lung cancer, nanomedicine, nanoparticles, VEGF, VEGFR

## Abstract

Lung cancer, the leading cause of cancer-related deaths worldwide, is characterized by its aggressive nature and poor prognosis. As traditional chemotherapy has the disadvantage of non-specificity, nanomedicine offers innovative approaches for targeted therapy, particularly through the development of nanoparticles that can deliver therapeutic agents directly to cancer cells, minimizing systemic toxicity and enhancing treatment efficacy. VEGF and VEGFR are shown to be responsible for activating different signaling cascades, which will ultimately enhance tumor development, angiogenesis, and metastasis. By inhibiting VEGF and VEGFR signaling pathways, these nanotherapeutics can effectively disrupt tumor angiogenesis and proliferation. This review highlights recent advancements in nanoparticle design, including lipid-based, polymeric, and inorganic nanoparticles, and their clinical implications in improving lung cancer outcomes, exploring the role of nanomedicine in lung cancer diagnoses and treatment.

## 1. Introduction

The latest data that the World Health Organization (WHO) published concerning lung cancer are worrisome: following breast cancer, lung cancer is the most common form of cancer diagnosed (1,350,000 new cases and 12.4% of the total new cancer cases) and the leading cause of death (1,180,000 deaths and 17.6% of total deaths due to cancer) in 2020, significantly increasing disease burden globally [1,2]. Since 1985, it has been the form of cancer with the highest incidence and mortality rate. Globally, the 5-year survival rate is 19%, but countries such as Japan (33%), Israel (27%), and the Republic of Korea (25%) report slightly increased survival rates [3]. Improvements in survival have been noted during the past decades, yet they do not match the survival gains accomplished in other cancer types. Also known as bronchogenic carcinoma, lung cancer is a malignant condition that occurs in either lung parenchyma or within the bronchi. Among the etiological factors responsible for the malignancy, tobacco smoking, secondhand tobacco smoke, occupational carcinogens, genetic predisposition, and gender are the most common carcinogens associated with lung cancer [4,5,6,7,8,9]. Based on the origin of cancer cells, two main forms have been described: small cell lung cancer (SCLC) and non-small cell lung cancer (NSCLC) [10]. A further classification of NSCLC exists, lung adenocarcinoma, squamous cell carcinoma, and large cell carcinoma, being underlined as individual conditions [5,11,12]. Moreover, an increase in adenocarcinoma incidence has been observed in the past decades, making it the most prevailing among all types of NSCLC. According to The American Cancer Society, an estimated 234,580 new cases of lung cancer have been diagnosed in the United States in 2024 (116,310 in men and 118,270 in women) and 85% of the cases are classified as NSCLC [13]. SCLC tends to have a more aggressive nature and a poorer prognosis, mainly because its rapid growth and spread. However, both forms display high lethality due to a lack of early diagnostic techniques. More than that, treatment of lung cancer is complex due to the lack of accessibility of the deeper lung regions when surgery is advised. Chemotherapy, on the other hand, has been proven to be a valuable tool for increasing the survival rate of those affected by this malignancy, but the main disadvantage besides the severe side effects is the nonspecific nature of the active pharmaceutical ingredient (API) delivery [14]. Nanomedicine has emerged as a promising avenue for both the diagnosis and treatment of SCLC and NSCLC, garnering significant interest from researchers. Its potential to revolutionize lung carcinoma management lies in its ability to offer a more controlled manner of treatment and enable an earlier detection. Researchers have been exploring various ways in which nanomedicine can enhance the control and treatment outcomes of lung cancer [15,16,17,18,19]. In this review, the existing literature is explored to uncover novel treatments for both SCLC and NSCLC, focusing on the application of nanomedicine. The aim is to provide a deeper understanding of the underlying molecular mechanisms of both forms of lung cancer, whilst examining how different nanosystems influence these mechanisms. Through this investigation, it is sought to shed light on the promising role of nanomedicine in improving the diagnosis and chemotherapeutic treatment of SCLC and NSCLC. Ultimately, it underscores the potential of nanomedicine to address the current challenges in lung cancer treatment, paving the way for more personalized, effective, and targeted therapeutic approaches.

## 2. Overexpression of VEGF and VEGFR

Angiogenesis is possible, among others, through interaction between VEGF and its complementary receptor, VEGFR (Figure 1). This enables continuous nutrition and oxygen supplementation towards cancerous cells of all types of tumoral tissue. Multiple forms of tumors have been correlated with excessive expressions of both VEGF and VEGFR, and treatment with antibodies against VEGF and inhibitors of VEGFR are popular in oncological departments [20]. Nevertheless, angiogenesis is linked to a high expression of VEGF, and has been identified in the maintenance of an inflammatory status and the further triggering of endothelial cells to produce protease and plasminogen activators [11]. Two types of receptors are described in the literature: tyrosine kinase (TK) receptors’ family with VEGFR-1 (feline McDonough sarcoma virus/fms-like tyrosine kinase-1, Flt-1), VEGFR-2 (fetal liver kinase-1, KDR/Flk-1), and VEGFR-3 (Flt-4) as representative members and neuropilin receptors, NRP-1 and NRP-2 being the two subtypes. The latter type acts rather as coreceptors as they add stability to the VEGF-VEGFR complex. Human endothelial factor also finds itself in numerous forms: VEGF-A, VEGF-B, VEGF-C, VEGF-D, VEGF-E, VEGF-F, placental growth factor (PLGF), and endocrine gland-derived VEGF (EG-VEGF) [21,22,23].

A literature review by Costache et al. [24] further explores how in different types of cancer, certain forms of VEGF and VEGFR are overexpressed (e.g., in NSCLC, the overexpression of VEGF-C predicts an unfavorable prognosis). Moreover, elevated levels of VEGF are linked with the confirmation of a tumoral tissue presence as many cells (existing vascular endothelial cells, cancerous cells, immune cells of the tumor microenvironment, and even precursors of endothelial cells) secrete it in the use of aberrant growth of the existing tumor [21,25,26,27]. Taking a closer look at the overexpression of VEGFR, it must be mentioned beforehand that even if the three representatives share similar structures, their activation and final response are different [28]. Receptor VEGFR-1 accepts VEGF-A, VEGF-B, and PIGF as ligands and through a phospholipase C (PLC-γ) signaling cascade contributes to angiogenesis, monocyte migration, and hematopoiesis. VEGFR-2, the most targeted one in oncotherapy, binds VEGF-A, VEGF-C, and VEGF-D and activates multiple signaling pathways that result in either cell proliferation, permeability of vessels, or the migration of cells (Figure 2).

Finally, VEGFR-3 also has affinity especially towards VEGF-C and VEGF-D and results in the spread of cancerous cells in lymph nodes from proximity and vasculogenesis [29,30,31]. Alternative splicing of VEGFR-1 pre-mRNA results in a soluble isoform of the receptor (sVEGFR-1), which is believed to control VEGFR-2’s activity. A similar soluble form of VEGFR-2 was also discovered, but its function remains unclear. In his published manuscript, Roskoski R [32] describes the general structure of VEGFR: three portions of the receptor are present. The extracellular portion consists of seven successive immunoglobulin-like domains (IgD1-IgD7); the first one, IgD1, also presents the N-terminal signal sequence. The transmembrane portion is inserted into the cell membrane, whilst the intracellular portion finds itself in the cytoplasm and consists of five domains: the juxtamembrane domain, proximal kinase domain, kinase insert domain, distal kinase domain, and C-terminal domain. Receptors are activated through binding between a key molecule and one of the IgDs, intracellular signaling through dimerization, and the autophosphorylation of the receptor, then enabling tumor capillarization [33]. As little [34] to no recent research explores the overexpression of VEGFR in a cancer context, papers by Meyer et al. [35] and Salameh et al. [36] were consulted as they experimented with how certain mutations to the encoding genes result in increased activity of the receptors. They found out that the activation segment of VEGFR-1 presents the amino acid asparagine in position 1050, whilst VEGFR-2 and VEGFR-3 present aspartate in the same position. Using wild-type and mutant variants of VEGFR-1 and VEGFR-2 (N1050D), they observed an increased autophosphorylation and activation of signaling cascades. An increase in the number of transmembrane receptors on cells’ surfaces can only be explained by similar mutations at the site of the respective encoding genes (Flt-1, Flk-1, and Flt-4), but research must be conducted toward this hypothesis.

## 3. Signaling Pathways in Lung Cancer

The VEGF/VEGFR pathway has been reported in SCLC [37,38] and NSCLC [39]. In both cancer forms, the pathway has angiogenic and non-angiogenic functions [40]. As far as NCSLC is involved, cancerous cells will secrete members of the VEGF family and, upon their binding to the corresponding receptors from the proximal vascular endothelial cells, several signaling cascades will activate. VEGF/VEGFR binding is responsible for activating the Ras/Raf/MEK/MAPK cascade, as shown in Figure 3. Rat sarcoma (Ras), a GTPase with weak activity, is released from its Ras/GDP form so it can activate both members of the mitogen-activated protein kinase (MEK) family (MEK1, MEK2) through an extracellular signal-regulated kinase (rapidly accelerated fibrosarcoma, Raf). This will further increase activity of extracellular signal-regulated kinases ERK1 and ERK2, mammalian homologs of mitogen-activated protein kinases (MAPKs) [41]. This particular pathway is responsible for the proliferation of endothelial cells [21,42].

Another signaling cascade starts with phospholipase C γ (PLC-γ) that further breaks down phosphatidyl-inositol-4,5-biphosphate (PIP2) present in the endothelial cell membrane into inositol triphosphate (IP3) and diacylglycerol (DAG). IP3 will increase intracellular levels of Ca^2+^, resulting in inserting protein kinase C (PKC) in the cells’ membrane. DAG will then activate PKC. Vascular regulation and permeability are the main results. Activated VEGFR can also produce the phosphorylation of phosphoinositide 3-kinase (PIK3). As shown in Figure 3, binding of PIK3 with PIP2 will convert the latter into PIP3 (phosphatidyl-inositol-3,4,5-triphosphate). PIP3 further phosphorylates serine/threonine-specific protein kinase (AKT) that will mainly trigger the mammalian target of rapamycin (mTOR), protein kinase implicated in both physiological and pathological responses of cells [43]. AKT also activates endothelial nitric oxide synthase (eNOS) that releases nitric oxide (NO) from L-arginine.

The Nck/Fyn/Cdc42/p38MAPK/MAPKAPK2 pathway can generate a contractile force that enables migration on endothelial cells. In this pathway, after the binding of VEGF, its corresponding receptor dimerizes itself and its tyrosine residues from the cytoplasmic region autophosphorylate [44]. One of them is Tyr1214 and it is directly involved in the activation of stress-activated protein kinase 2 (SAPK2). The latter is a member of the p38 family of mitogen-activated protein kinases and has multiple clinical implications such as the one at the site of inflammation [45]. It was shown that cell division control protein 38 (Cdc38), a small GTPase part of the Rho family, is an essential intermediate between the autophosphorylation of the Tyr1214 site of the VEGFR and the activation of the SAPK2-p38 module (MAP3K, MAP2K, SAPK2/p38) that further triggers the activation of MAPKAP kinase 2/3 initiating actin remodeling and actin-based motility of endothelial cells, as shown in Figure 3 [46]. Only a few years later, implications of the adapter non-catalytic region of tyrosine kinase adaptor protein 1 (Nck) and of the Src kinase family member Fyn were described in this particular pathway. It is important to mention that all these signaling cascades entangle with one another (e.g., the implication of PIP2 in the first and third signaling pathways described) to better the angiogenesis response in favor of the tumor. On the other hand, in SCLC, a review by Yuan et al. [47] revealed only one common signaling pathway (the PIK/AKT/mTOR pathway) that is relevant as far as therapeutic strategies for treating lung cancer are involved.

## 4. Receptor-Based Targeting

Receptor-based targeting seems to be an attractive alternative of administering therapeutics as it has the potential to improve the efficacy of encapsulated drugs and, when it comes to cancer, it could minimize drug-associated side effects. As such, the surface engineering of nanocarriers seems to be of utmost importance in the current nanotechnological scenario. There are several receptors that are overexpressed on the surface of cancer cells, such as transferrin, lactoferrin, lectin, folate, human EGF receptors, scavenger receptors, nuclear receptors, integrin, etc. [48]. CXCR4 is such a receptor as it is predominantly localized in the plasma membrane of tumor cells. C-X-C chemokine receptor-4 (CXCR4a) is a transmembrane G-protein-coupled receptor (GPCR) classified as a member of the family I GPCR or rhodopsin-like GPCR family [49,50,51,52]. The natural ligand for CXCR4 is chemokine stromal cell-derived factor-1 (SDF-1 or CXCL 12), an 8 kDa, 67-residue CXC chemokine peptide, originally isolated from a bone marrow stromal cell line, and it is the natural ligand for CXCR4 [49,53]. The evaluation of the CXCR4 expression in small cell lung cancers showed that it was expressed with high intensity in almost all of the SCLC samples. The activation of the receptor leads to the proliferation of cells and chemotaxis towards the source of the ligand [54,55]. As such, the CXCR4/CXCL12 axis is important for the homing and retention of stem cells and the trafficking of lymphocytes towards the sites of tissue damage or inflammation [54,56,57]. The involvement of CXC receptor-4 in tumor progression, metastasis, adaptation to hypoxia, and stem cell survival is implied in many studies, which demonstrate the tie between CXCR4 expression and tumor aggressiveness when it comes to metastatic spread, which often leads to limited patient overall survival. Furthermore, the expression of CXCR4 by cancer cells seems to be associated with malignancy potential and tumor recurrence. Therefore, the CXCR4/CXCL12 axis seems to be a promising target for cancer therapies [54]. The potential of CXCR4 antagonists as imaging reagents has been highlighted in multiple studies. One such treatment is the one with AMD3100 labeled with Gallium-68 in breast cancer-bearing mice. Plerixafor (AMD3100) has been approved by the United States Food and Drug Administration and in the European Union by the European Medicines Agency for stem cell mobilization [54,58,59]. The 68Ga-labeled high-affinity CXCR4 ligand (68Ga-CPCR4-2/cyclo(D-Tyr(1)-[NMe]-D-Orn(2)-[4-(aminomethyl) benzoic acid) seems to be characterized by high in vivo stability and distinct and specific tumor accumulation [54,60]. Several animal trials showcase the in vivo anti-metastatic efficacy of CXCR4 antagonists, such as TF 14016, CTCE9908, and AMD3465.

PARP inhibitors seem to be an attractive solution when it comes to targeted therapy of SCLC. Compared to other lung cancer subtypes and normal lung epithelial cells, SCLC shows a high-PARP-expression profile. PARP enzymes are a group of proteins involved in DNA break recognition and repair, chromatin remodeling, and the regulation of transcription [61]. PARP inhibitors may also have the potential to enhance the cytotoxic response to chemotherapy, ionizing radiation, and even the newly introduced immunotherapy in SCLC [62]. However, because these suspicions have not yet been confirmed by clinical trials, predictive biomarkers are required to maximize the clinical efficacy of PARP inhibitors. An example of such a biomarker is SLFN11, strongly associated with veliparib efficiency (veliparib is a poly adenosine diphosphate ribose polymerase (PARP)-1 and -2 inhibitor) [63]. Even though the use of PARP inhibitors has an important drawback through their additive toxicity profile when administered together with cytotoxic therapy, veliparib is the most tolerable in combination with chemotherapy, as it shows the lowest amount of trapping PARP. Olaparib, rucaparib, and niraparib show a moderate PARP trapping and talazoparib shows the highest degree of trapping [64]. It is important to note that any adverse events were clinically manageable and did not affect the delivery of standard chemotherapy [65]. Another alternative may be AURKA inhibitors for SCLC with high MYC expression [66]. Aurora-kinase A (AURKA) is part of the Aurora family of serine/threonine kinase and regulates mitosis and maintains genetic fidelity [67]. AURK inhibition combined with chemotherapy strongly suppresses proliferation and tumor growth both in vitro and in vivo. Several preclinical studies have pointed towards the benefits of the use of inhibitors against the c-Kit receptor, EGFR, insulin-like growth factor receptor, and c-MET receptor tyrosine kinases in SCLC. However, subsequent clinical trials did not show statistical significance when it comes to the survival benefits [63,64,66,67,68]. DLL3 [69] and BLC-2 [70] are also frequently expressed on the surface of SCLC tumor cells. However, clinical trials showed that patients who received treatments with DLL3 or BLC-2 inhibitors had considerably higher rates of toxicities without a clear survival benefit [71,72,73].

Cisplatin and gemcitabine (GEM) are common clinical chemotherapeutic agents when treating NSCLC. GEM inhibits nucleoside enzymes and Cisplatin induces DNA damage. However, their usage in the clinical field pointed towards serious side effects mainly attributed to their deficiency in selectivity [74]. An alternative could be ultra-small platinum nanoparticles (USPtNs), which could leak Pt ions under acidic conditions, such as in cell endosomes or lysosomes. Highly toxic Pt ions released from the dissolved USPtN interface have been found to trigger corrosion-activated and Cisplatin-like chemotherapeutic functions. Therefore, efficient cancer therapy is expected to be achieved through targeted delivery of USPtNs. However, this approach has many drawbacks, such as a poor tumor-specific accumulation, short half-life, and broad range of toxicity to normal tissues, which significantly limit the use of USPtNs in vivo [75]. Some NSCLCs have suffered EGFR-sensitizing mutations. As such, EGFR tyrosine kinase inhibitors (TKIs) may be a great alternative when it comes to targeting NSCLC. In the metastatic setting, EGFR TKIs have been established as first-line therapy because of their progression-free survival (PFS) benefit and excellent tolerability [76,77,78,79]. Although initially effective, resistance to EGFR TKI therapy does emerge. Therefore, combined VEGF and EGFR inhibition represents a rational combination strategy for EGFR-mutant NSCLC treatment as VEGF and EGF share common downstream signaling pathways and may function exclusively from one another during oncogenesis and acquired therapeutic resistance [76].

It is safe to say that the inflammatory responses of lung cancer are obtained through the immune system, which involves chemokines (a family of chemotactic cytokines) as regulators of immune cell trafficking in the body. Chemokines interact with seven transmembrane-G-protein-coupled receptors [80,81]. The repertoires of chemokine receptors, which guide their trafficking, retention, and function in target organs, are specific for every immune cell subtype [80,82]. The activation of the chemokine/chemokine receptor axis within tumors induces autocrine and paracrine loops promoting tumor growth and angiogenesis, which, in turn, leads to anti-tumor immune responses. Such loops are involved in non-small cell lung cancer (NSCLC) carcinogenesis [80]. Different cytokine and chemokine/chemokine receptor complexes characterize distinct types of immune responses, each being related to another type of cancer. Several mutations regarding driver genes such as epidermal growth factor receptor (EGFR), anaplastic lymphoma kinase (ALK), ROS1 protooncogene receptor tyrosine kinase (ROS1), and serine/threonine protein kinase v-Raf murine sarcoma viral oncogene homolog B (CRAF) were discovered in NSCLC, which increased treatment options against NSCLC significantly [83]. The family of epidermal growth factor receptor tyrosine kinase (ErbBs) is an essential component of the cellular signaling pathways that control vital processes such as cell survival, differentiation, proliferation, and apoptosis [84,85]. The source of the ErbB family’s name is the erythroblastic leukemia viral oncogene, for which the receptors are identical. The four structurally conserved members of this family are epidermal growth factor receptors EGFR/ErbB1, ErbB2, ErbB3, and ErbB4. The common domain structure includes an intracellular area with a juxtamembrane domain, tyrosine kinase (TK) domain, and C-terminal tyrosine-rich region. The ligands bind to the extracellular domain, which includes a hydrophobic transmembrane segment and an extracellular segment. The growth factors (transforming growth factor-alpha (TGF-alpha) and epidermal growth factor (EGF)) bind to the extracellular portion of the receptor. After the ligand/receptor complex is created, this results in the activation and phosphorylation of the TK domain at its C-terminal residues, which causes the receptor to homo- and/or heterodimerize and initiate downstream signaling cascades. EGFR is overexpressed in approximately 60% of individuals with NSCLC [84]. To treat EGFR-active mutations, therapy with EGFR tyrosine kinase inhibitors (TKIs) seems intuitive. However, the majority of patients become resistant to these inhibitors after 10 to 12 months of initial therapy (due to the acquisition of a second-site mutation (T790 M)) and so, several third-generation EGFRTKIs, such as Osimertinib, are engineered to overcome drug resistance. Cetuximab (CET), an immunoglobulin G monoclonal 38 antibody, could be an alternative when it comes to targeting EGFR overexpressing cancer cells via 39 receptor-mediated EGFR phosphorylation and signal shutdown. CET is used either alone or in combination with other drugs, which increase its targeting ability, in the treatment of advanced NSCLC [86].

## 5. Nanocarrier-Mediated Drug Delivery Systems

The standard treatment for small cell lung cancer (SCLC) and for non-small cell lung cancer (NSCLC) is represented by surgery accompanied by radio- and chemotherapy. However, numerous advances in treatments over the past decade have revolutionized the way this incurable disease is approached. Of particular interest are the incorporation of immunotherapy and targeted therapy. Nanosystems seem to be an attractive solution to different types of cancer nowadays, including both SCLC and NSCLC, as they offer the possibility of targeted treatment and are relatively easy to manufacture by manipulating their structure, geometry, materials, and surface chemistry (Table 1, Figure 4. They can also be produced based on tumor genetic profiles, which results in a more effective drug choice for personalized patient treatment. Various characteristics of nanosystems make them such appealing treatments. Size: Perfectly sized nanosystems can be obtained via engineering deformability so that they are small enough to pass through the sinusoids of the spleen but large enough to avoid accumulating in the liver. As such, a prolonged circulation of the treatment can be potentially obtained [57]. Surface chemistry: The chemical composition of the nanoparticles fulfills two main roles: it affects opsonization and it can target cells or organelles by attaching to ligands on the surface of the engineered nanosystems, as the ligands bind to receptors overly expressed on the surface of rapidly dividing cancerous cells. Stimuli-responsive release of cargo: Certain nanomaterials can respond to internal or external stimuli. The pH in the tumor tissue and late endosomes and lysosomes is lower than that of healthy tissues, which may trigger the breaking of certain chemical bonds, unstable under acidic conditions. Subsequently, the cargo is selectively released into the tumoral cell of interest. Other stimuli of interest include a light, magnetic field; ultrasound; etc. Camouflage: Biomimetic nanomaterials offer numerous advantages such as prolonged circulation, cell-specific targeting, immune escape, lower toxicity, and better biocompatibility [87].

### 5.1. Liposomes

Liposomes (Figure 5A) are lipid-based nanoparticles that are used in current practice for delivering chemotherapeutic agents such as Cisplatin (in its liposomal encapsulated form, Lipoplatin^TM^) due to their various advantages. Liposomes add stability, solubility, and pharmacokinetics to the pharmaceuticals they are carrying, significantly improving their tissue penetration through the enhanced permeability and retention (EPR) mechanism. These nanocarriers also exhibit a non-toxic, biodegradable, and biocompatible nature, reducing the systematic toxicity associated with the free drug [13,94]. They are used for active, passive, pH, magnetic, stimuli-responsive, and even thermo-responsive targeting [1], making liposomes one of the most successful drug carrier systems. Liposomes are spherical vesicles formed from a lipid bilayer and an aqueous inner cavity with particle sizes ranging from 25 to 2500 nm [95]. Liposomes can be classified according to their functional modifications into conventional, polyethylene glycol-glycated (PEGylated), ligand targeting, and theranostic ones. Conventional liposomes can be charged either positively or negatively by inserting cationic or anionic phospholipids, respectively, in the lipid bilayer and they can carry both hydrophilic and lipophilic drugs: the first ones are entrapped in the aqueous inner cavity of the liposome and are gently released without interfering with the stability of the nanocarrier, whilst the latter ones are captured into the lipid bilayer and can affect stability. Other disadvantages can also be outlined. For example, conventional liposomes are rapidly eliminated through phagocytosis of macrophages [1,96]. PEGylated liposomes contain surface-grafted lipid derivates conjugated with polyethylene glycol that enable them to be avoided by the reticuloendothelial system, ensuring a longer circulation in the blood [97]. In ligand-targeting liposomes, targeting molecules such as monoclonal antibodies, proteins, growth factors, glycoproteins, and carbohydrates are conjugated on the liposomes’ surface, enabling an active manner of targeting by using specific pathological changes in the tumor environment (e.g., overexpression of proteins) [98]. Theranostic liposomes have targeted ligands inserted on the lipidic bilayer while containing imaging agents within their core, making diagnoses possible through this particular nanocarrier system, but hybrids in which chemotherapeutical wells are encapsulated in either the core or bilayer have also been described. Theranostic formulations have the potential to provide valuable information on the target site and off-target accumulation of pharmacologically active agents, a more controlled therapy being provided [99,100]. Switching to a clinical setting, the very first FDA (Food and Drug Administration)-approved nanoparticle-based drug delivery system was DoxilTM, a stealth TMPGEylated liposomal doxorubicin still in usage today that has shown significant results in the treatment of SCLC when combined with standard therapies and additional growth factors [101]. In a comprehensive review by Alshammari et al. [14], multiple clinical trials that use liposome-mediated drugs for lung carcinoma were described. Liposomes loaded with Irinotecan are currently being tested in NSCLC, whilst parenteral liposomal Irinotecan in combination with other chemotherapeutics such as niraparib is being researched in clinical trial settings for both SCLC and NSCLC. Liposomes embedded with Triptolide are also being tested for NSCLC. Phase II and III clinical trials are also exploring Irinotecan hydrochloride and Topotecan in liposome nanocarriers to explore the duration of response and overall survival in patients with lung carcinoma. Another review by Abdulbaqi et al. [102] summarizes clinical trials that have been conducted on inhalable anticancer drug-loaded lipid-based nanocarriers for the treatment of lung cancer, a novel concept in the recent literature. A single phase I clinical trial that used Cisplatin dipalmitoyl phosphatidylcholine and cholesterol liposomes on 16 subjects with NCSLC and 1 with SCLC is mentioned, with promising results despite the mild adverse effects [103]. Liposomes are described in the literature to be useful in therapies that target the downregulation of VEGF. Zhang et al. manufactured a live drug carrier, paclitaxel-in-liposomes-in-bacteria, which showed a quicker drug delivery and therefore, a remarkable inhibition of lung cell proliferation, amongst the downregulation of VEGF, showing a promising new therapy for lung cancer management [104]. In 2020, Zhang et al. described a tripeptide lipid nanoparticle loaded with paclitaxel and anti-VEGF siRNA that successfully delivered the medicinal cargo to lung cancer cells, leading to an important anti-tumor effect, by inhibiting VEGF expression and inducing apoptosis [105], underscoring the significant role of liposomes in lung cancer management, particularly in therapies targeting the downregulation of VEGF. By enhancing drug delivery efficiency and promoting anti-tumor effects, such as the inhibition of VEGF expression and the induction of apoptosis, liposome-based systems present a promising approach for improving lung cancer treatment outcomes.

### 5.2. Micelles

Micelles (Figure 5B), akin to liposomes, lipidic nanocapsules, and nanostructured lipid nanocarriers [1], are lipid-based nanocarriers that were first used as drug vehicles in the 1980s. Their size is around 50 nanometers and they have a structure consisting in a drug-loaded hydrophobic core and a polyethylene glycol hydrophilic shell. They can carry and protect insoluble hydrophobic medication and have remarkable tissue penetration and cellular uptake mainly due to their size. However, inconsistent stability and premature drug release have been noticed as the main disadvantages. In NSCLC, PLGA–PEG–maleimide micelles and cremophor-free paclitaxel-loaded PLGA-b-methoxy PEG polymeric micelles loaded with Docetaxel and Cisplatin, respectively, have been explored, with results turning out to be promising [106,107]. In their manuscript, Wan et al. [108] describe in animal models of SCLC and NSCLC how a particular polymeric micelle system based on two different copolymers within the amphiphilic block in combination with alkylated Cisplatin showed a superior anti-tumor activity. In a 2020 study, Chang et al. describe the manufacturing of doxorubicin-loaded micelles. The authors concluded that doxorubicin-loaded micelles targeted with anti-VEGF antibodies show a higher anti-tumor effect than the non-targeted micelles [109]. Also, complex micelles have been used to deliver siRNA to silence the VEGF gene. Kanazawa et al. describe a nanosystem based on micelles, which can enhance the stability of siRNA (siVEGF), so it can target and be delivered to the cancer cells that present an increased level of VEGF [110].

### 5.3. Polymeric Nanoparticles

Polymeric nanoparticles measure between 1 and 1000 nm and display valuable characteristics such as non-toxicity, non-immunogenicity, biocompatibility, and biodegradability. One of the major advantages of these nanocarriers is that they can be easily fabricated, with solvent evaporation and diffusion, reverse salting, and nanoprecipitation being among the methods currently used. Polymeric nanoparticles were previously classified into nanocapsules (reservoir systems) and nanospheres (matrix systems) [111]. In terms of structure, polymeric nanoparticles (nanospheres) describe a lipid–PEG outer layer on which multiple targeting agents can be inserted, whilst the inner core is available for encapsulating the active drug in a polymeric matrix [112]. Both FDA and EMA (European Medicines Agency) offered approval for the usage of the nanocarrier in clinical contexts [113]. Phase II clinical trials that used PEG–poly(lactic acid) (PLA) block polymers to create polymeric nanoparticles encapsulated with paclitaxel and Cisplatin (Genexol-PM) for the treatment of NSCLC have been described [107]. Other clinical trials explored the usage of Abraxane (Celgene) for advanced NSCLC [114]. In their original research article, Arslan et al. [115] were successful in proving how PEG–poly(lactide-co-glycolide) (PLGA) polymeric nanoparticles (Figure 5D) loaded with Irinotecan and a STAT3 inhibitor (Stattic) decreased side effects and displayed effective anti-tumor activity in animal models suffering from SCLC, the pathway being suspended in both in vitro and in vivo contexts. Another study [116] also explored Silibin-loaded poly(caprolactone)/pluronic F68 inhalable nanoparticles for both SCLC and NSCLC, proving a sustained release of the drug in the systemic circulation, inhibition of the tumor growth, and better efficacy of the drug.

### 5.4. Dendrimers

Dendrimers (Figure 5C) are also known under the name of arborols or cascade molecules. Their size, 1 to 100 nm, enables them to have a high permeation and circulation and, compared to liposomes or micelles, they exhibit more stability in unfavorable conditions such as temperature and pressure [117,118]. Dendrimers are intensely branched and uniformly structured: from the inner core, numerous complex extensions expand in a three-dimensional manner to form cavities in which the active pharmaceutical ingredient (API) will be trapped and further delivered at the tumor site. A maximum amount of API can be delivered to target through dendrimers. A functionalized group corona made up from surface end groups of the branches facilitates the manipulation of the nanocarrier so both active and passive targeting can be made possible [119,120]. A classification would underline high-molecular-weight dendrimers that have hyperbranched, dendronized, and brush polymers attached to the core, and low-molecular-weight dendrimers in which the constituting polymers are monodispersed and highly symmetrical [121]. Both have the capacity for adjustments of their size, form, and surface, making dendrimers particularly versatile as nanocarrier systems [122]. According to the environment, dendrimers will be able to overcome the first-pass effect and avoid immunity mechanisms and the penetration of healthy cells, off-target interactions also being minimal [123]. Multiple layers of branches can be manufactured, enabling a classification by generation: To the inner core, generation 0 (G0) and first-generation (G1) monomers will be added, making possible the linkage of second-generation (G2) monomers. Further arrangement with G3 and then with G4 monomers can be possible [124]. The following drugs have been successfully entrapped into dendrimers to facilitate lung cancer therapy: Cisplatin [125], Doxorubin [126], EndoNt [127], and doxorubicin [128]. Cisplatin was heavily encapsulated in poly(amidoamine) (PAMAM) dendrimers through sonication and centrifugation and results showed a minimum loss of the API at targeted sites, followed by a high release rate, good biocompatibility, and inhibition of tumor cells’ activity. Doxorubin with doxorubicin-conjugated PAMAM dendrimers was explored in the context of lung metastasis in animal models, a higher number of lung nodules being reduced compared to therapy with the free form of Doxorubin. EntoNt, successfully carried to the tumor site by PAMAM dendrimers, demonstrated stable biocompatibility and efficient capturing on SCLC cells that were immobilized or under flow, respectively. Finally, doxorubicin encapsulated in PEG–poly(L-lysine) (PLL) dendrimers proved prolonged action and lower toxicity related to the lungs compared to the equivalent dose of Doxorubin. Furthermore, dendrimers loaded with bromoenol lactone inhibitors also showed a higher therapeutic index [1].

### 5.5. Quantum Dots

Quantum dots are colloidal particles with a 2 to 100 nm size that display an atom-like behavior attributed to their electronic activity (Figure 5E). The structure of quantum dots consists of a semiconductor metal core that is trapped into a capping shell covered in a coat of polymers. This structure is adapted for trapping charge carriers in a volume somewhat equal to the quantum mechanical wavelength of its components [1,112]. On the hydrophilic surface of the latter, different molecules can be conjugated [129]. Some of the chemical elements used in the core manufacturing are cadmium selenide (CdSe), cadmium tellurite (CdTe), zinc sulfide (ZnS), zinc selenide (ZnSe), gallium nitride (GaN), gallium arsenide (GaAs), indium phosphide (InP), and indium arsenide (InAs) [130]. Due to the restricted wavelength of the emission spectrum, these nanocarriers display unique photo-physical characteristics (large absorption spectra, high photobleaching, photostability, high fluorescence emission) that make them become eligible for clinical imaging in cancer contexts [131]. Quantum dots also hold great promise for the treatment of different carcinomas, the lung included: the modifiable nature of the surface of quantum dots can ensure solubility and biocompatibility, and their size offers remarkable permeability at the tumor site, enhancing the effectiveness of targeting [132,133]. Coating these nanoparticles with lipids has been shown to increase solubility even more [134]. The existing literature explored quantum dots as nanocarriers for doxorubicin against folate receptors [135] and overexpressed protein CD44 [136], as well as for combinations of paclitaxel, doxorubicin, and Carboplatin against Bcl-2 siRNA [112], results showing high cytotoxicity against lung cancer cells. Moreover, a review by Tade et al. [137] explored how graphene quantum dots have great potential for both diagnostic and curative applications in the carcinoma of the lungs, including several in vivo and in vitro studies with promising results. Ngema et al. developed a targeted nanosystem for paclitaxel delivery using superparamagnetic iron oxide nanoparticles (SPIONs) functionalized with anti-VEGF peptides. They showed a 76.6% tumor regression in a lung tumor xenograft mouse model, concluding that it could provide a potential effective treatment [138].

**Figure 4 ijms-25-11235-f004:**
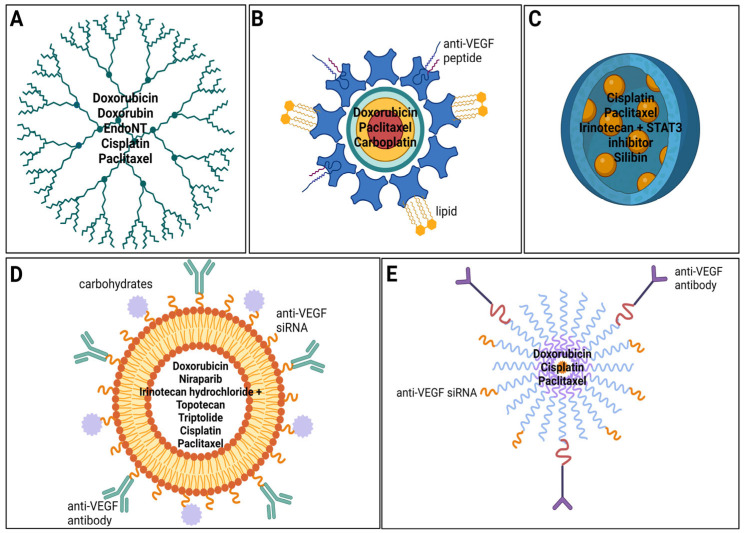
Schematic representation of nanocarrier systems used in lung cancer treatment. (**A**) Dendrimers. (**B**) Quantum dots. (**C**) Polymeric nanoparticles. (**D**) Liposomes. (**E**) Micelles. Images created with BioRender.com (accessed on 15 October 2024).

**Figure 5 ijms-25-11235-f005:**
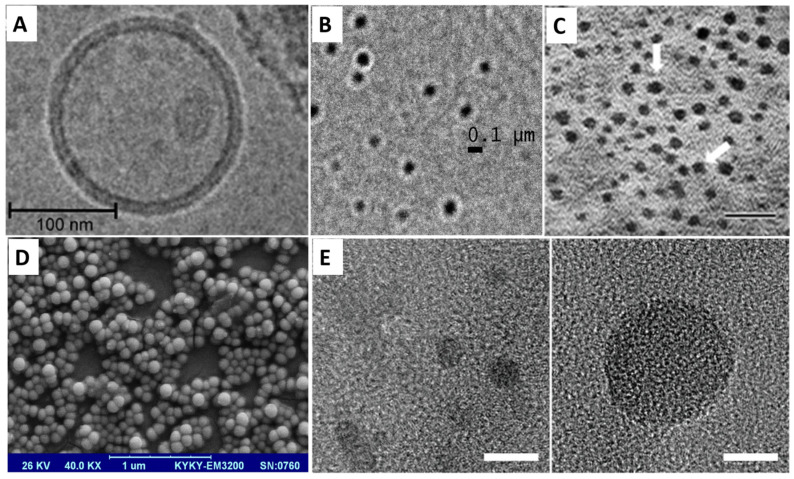
(**A**) Liposome in TEM. Image adapted from Kotouček et al. [139]. (**B**) Micelles in TEM. Image adapted from Nair et al. [140]. (**C**) Dendrimers (scale bar 4 nm). Image adapted from Walworth et al. [141]. (**D**) PEG–poly(lactic acid) nanoparticles. Image adapted from Ghasemi et al. [142]. (**E**) Quantum dots. (scale bar: 10 nm) Image adapted from Tachi et al. [143].

## 6. Conclusions

In conclusion, the application of nanomedicine in the treatment of SCLC and NSCLC has proven to be a promising approach with significant potential for improving patient outcomes. Our review explored various aspects of lung adenocarcinoma, including the overexpression of VEGF and its corresponding receptor (VEGFR), the intricate signaling pathways involved in lung cancer progression, and nanomedicine applications to it. Furthermore, we discussed the role of nanoscale drug delivery systems, including nanocarrier-mediated drug delivery and receptor-based targeting strategies, in enhancing the efficacy and specificity of lung cancer treatment. These innovative approaches offer a more controlled and targeted delivery of therapeutics, potentially minimizing off-target effects and improving patient response to treatment. Overall, this review highlights the promising role of nanomedicine in revolutionizing the diagnosis and treatment of both SCLC and NSCLC. By gaining a deeper understanding of the underlying molecular mechanisms and leveraging advanced nanoscale delivery systems, we are moving closer to more effective and personalized therapeutic strategies for lung cancer patients.

## Figures and Tables

**Figure 1 ijms-25-11235-f001:**
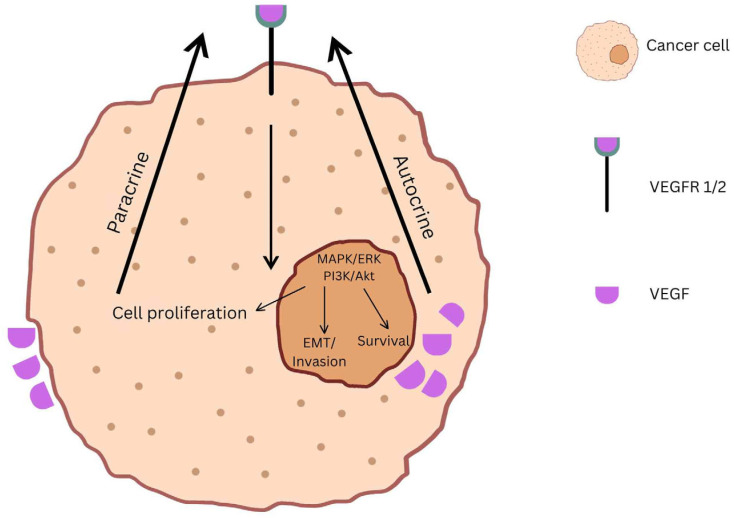
Autocrine and paracrine VEGF signaling in cancer cell proliferation and survival.

**Figure 2 ijms-25-11235-f002:**
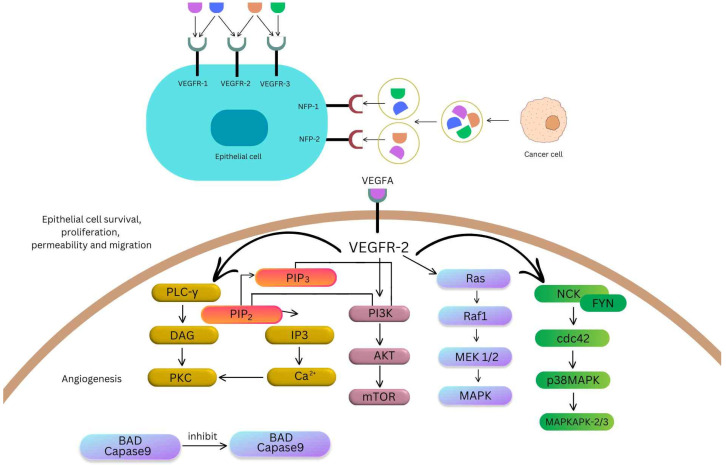
VEGFR-mediated signaling pathways promoting angiogenesis, cell survival, and migration in cancer.

**Figure 3 ijms-25-11235-f003:**
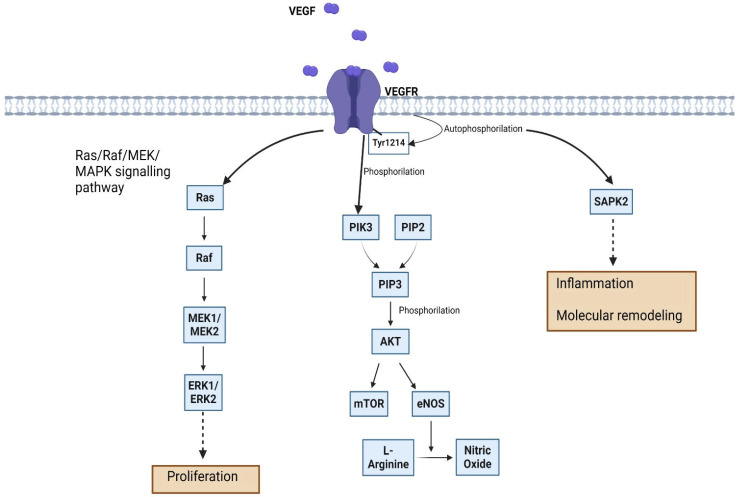
VEGF and VEGFR signaling pathways involved in lung cancer development. Image created with BioRender.com (accessed on 31 July 2024).

**Table 1 ijms-25-11235-t001:** Comparison of Nanoparticle-Based Drug Delivery Systems: Composition, Functionalization, Therapeutic Agents, Advantages, and Disadvantages.

Type of Drug Delivery System	Composition	Therapeutic Agents Carried	Functionalization	Advantages	Disadvantages	Ref.
Liposomes	Spherical vesicles formed from lipid bilayer and aqueous inner cavity Particle sizes: from 25 to 2500 nm	Both hydrophilic and lipophilic drugs	PEGylated antibody functionalization Protein functionalization Growth factor functionalization Glycoprotein and carbohydrate functionalization	Stable Soluble Increases tissue permeability Non-toxic Biocompatible	Easily eliminated	[88]
Micelles	Spherical vesicles formed from hydrophobic core and polyethylene glycol hydrophilic shell Particle sizes: 50 nm	Insoluble hydrophobic therapeutic agents	Hydrophilic polymer functionalization	High tissue permeability High cellular uptake	Inconsistent stability Premature drug release	[89]
Polymeric nanoparticles	Nanoparticles with lipid–PEG outer layer and polymeric matrix. Particle sizes: between 1 and 1000 nm	Mostly hydrophilic drugs	PEGylated antibody functionalization Protein functionalization Growth factor functionalization Glycoprotein and carbohydrate functionalization	Non-toxic Non-immunogenic Biocompatible Biodegradable Easily fabricated	Some residues from preparation technique might interact with drug delivery	[88,90]
Dendrimers	Highly branched nanopolymeric structures Particles sizes: 1 to 100 nm	Hydrophobic drugs	Organic and inorganic molecule functionalization	Stable Multiple-drug entrapment Controlled delivery	Mostly show low solubility in aqueous solutions	[91]
Quantum dots	Colloidal particles with semiconductor metal core and capping shell covered in coat of polymer Particles sizes: 2 to 100 nm	Hydrophobic drugs Mostly used in diagnosis	Lipid functionalization Amine functionalization Protein functionalization DNA functionalization	Electrochemical fluorescence emission properties	Hydrophobic if not functionalized	[89,92,93]

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
