# Peer review of "Emerging Nanomedicine Approaches in Targeted Lung Cancer Treatment"

_ijms, 2024, doi:10.3390/ijms252011235_

Round 1
Reviewer 1 Report
Comments and Suggestions for Authors
This manuscript presents the development of nanoparticles which can deliver therapeutic agents to the cancer cells or treat, particularly lung cancer cells, with relative less toxicity. The authors have reviewed numerous previously published papers; however, more recent studies should be covered in this review article. Therefore, this manuscript needs major revision prior to publication. Detailed comments are provided below.
Comments:
1. What is the different from previous review papers? Although the authors cited recent studies, there are already many similar works on this topic. A search of “PubMed” using the keyword as “nanoparticle and lung cancer” yields 342 review papers published within last 5 years. Therefore, the manuscript should offer a unique perspective or recent findings which could be distinguished from previous review papers.
2. This review manuscript contains 124 references, but only 45 references (about 36%) were published from the last 5 years. Since many similar papers are existed, the authors should focus on the latest findings and provide a comprehensive review of the most recent researches.
3. Also, the authors should ensure that all references contain detailed information, such as journal name, year, volume #, or page #.
4. The authors summarized the inhibitors against autophagy, however, they need to present more information of those inhibitors, including chemical structures.
5. Please provide more figures and tables of summary. With figures and tables, readers can understand the content of this review manuscript more easily.
6. English proof reading is required to fix multiple typos and spacing errors.
Comments on the Quality of English LanguageEnglish proof reading is required to fix multiple typos and spacing errors.
Author Response
This manuscript presents the development of nanoparticles which can deliver therapeutic agents to the cancer cells or treat, particularly lung cancer cells, with relative less toxicity. The authors have reviewed numerous previously published papers; however, more recent studies should be covered in this review article. Therefore, this manuscript needs major revision prior to publication. Detailed comments are provided below.
First of all, we would like to thank you for your time and valuable feedback in reviewing our manuscript. We appreciate your insights, and we acknowledge the importance of incorporating more recent studies to ensure the review is comprehensive and up-to-date. Below, we address your detailed comments and outline the revisions made to the manuscript.
Response to Reviewer Comments
Comment 1:
What is different from previous review papers? Although the authors cited recent studies, there are already many similar works on this topic. A search of “PubMed” using the keyword “nanoparticle and lung cancer” yields 342 review papers published within the last 5 years. Therefore, the manuscript should offer a unique perspective or recent findings which could be distinguished from previous review papers.
Response:
Thank you for your feedback. The manuscript presents several unique perspectives and findings that distinguish it from previous review papers on nanomedicine and lung cancer:
- Focus on Specific Types of Lung Cancer: This review specifically addresses both small cell lung cancer (SCLC) and non-small cell lung cancer (NSCLC), exploring their distinct molecular mechanisms and treatment challenges, which may not be comprehensively covered in other reviews.
- Recent Advances in Nanoparticle Design: It highlights the latest advancements in nanoparticle design, including lipid-based, polymeric, and inorganic nanoparticles, with an emphasis on their clinical implications for improving lung cancer outcomes.
- Innovative Drug Delivery Systems: The review delves into nanoscale drug delivery systems, such as polymeric nanoparticles and liposomes, that enhance the efficacy and specificity of lung cancer treatments. It also provides detailed insights into their mechanisms and clinical trial results, which may not be extensively covered in other literature.
- Theranostic Approaches: The manuscript explores theranostic formulations that combine therapeutic and diagnostic capabilities, offering more controlled therapies and potentially improving patient management.
Comment 2:
This review manuscript contains 124 references, but only 45 references (about 36%) were published within the last 5 years. Since many similar papers exist, the authors should focus on the latest findings and provide a comprehensive review of the most recent research.
Response:
Thank you for your valuable feedback. We have updated the manuscript to include additional recent studies. However, we would like to highlight that many foundational and highly relevant studies in nanomedicine are older than five years, and these remain crucial to understanding current advancements in the field.
Comment 3:
The authors should ensure that all references contain detailed information, such as journal name, year, volume, and page numbers.
Response:
Thank you for your observation. We have made the necessary adjustments to ensure all references contain complete information.
Comment 4:
The authors summarized the inhibitors against autophagy, but they need to present more information about those inhibitors, including their chemical structures.
Response:
Thank you for the suggestion. We have included more detailed information about the inhibitors, including their chemical structures, to enhance the content.
Comment 5:
Please provide more figures and tables for summary. With figures and tables, readers can understand the content of this review manuscript more easily.
Response:
We appreciate the suggestion. Additional figures and tables have been added to the manuscript to improve clarity and comprehension.
Comment 6:
English proofreading is required to fix multiple typos and spacing errors.
Response:
We have carefully proofread the manuscript and corrected all identified typos and spacing errors.
Reviewer 2 Report
Comments and Suggestions for Authors
In this article, Alexandru et al. reviews the advancements in nanoparticle drug delivery systems for lung cancer. The manuscript includes up-to-date references and the topic is relevant. However, it presents several major flaws that deem the manuscript unsuitable for publication in International Journal of Molecular Sciences. The authors should consider the following suggestions:
1) The English needs to be extensively improved. The authors often use unnecessarily short phrases, and change the tenses (past, present and future) and pronouns (1st and 3rd person).
2) The manuscript lacks images of relevant results on drug delivery systems for lung cancer.
3) The authors should include tables that summarize the current drug delivery systems for lung cancer therapy, including the composition, therapeutic agents, functionalization, advantages and disadvantages.
4) Several sections of the paper mention findings and clinical trials without properly citing the respective article. For example, see the section from line 192 to 228.
5) The authors should reconsider the naming of chapter 4, as it includes a mixture of information related with nanosystems, targeting strategies and drugs that would be better included in other chapters. For instance, all the discussion on targeting should be in chapter 5, while the discussion on commonly used drugs would be more adequate for a different chapter.
6) Lines 283 to 286 seem to belong to the introduction.
7) Despite the authors highlighting the VEGF and VEGFR with an extensive description of their overexpression, the manuscript misses a discussion on nanosystems that explore the targeting of VEGF/VEGFR.
8) Considering the topic of the review, a larger discussion on the nanosystems would be expected. However, in each nanosystem subchapter, only few studies are mentioned, not critical discussed, and mostly lack any relation with the topic. Besides, unclear phrases and lack of citations are commonly found as in the case of lines 439 to 448.
Comments on the Quality of English LanguageThe authors often use unnecessarily short phrases and change the tenses (past, present and future) and pronouns (1st and 3rd person). Besides, the manuscript displays sections with difficult-to-read phrases.
Author Response
In this article, Alexandru et al. reviews the advancements in nanoparticle drug delivery systems for lung cancer. The manuscript includes up-to-date references and the topic is relevant. However, it presents several major flaws that deem the manuscript unsuitable for publication in International Journal of Molecular Sciences. The authors should consider the following suggestions:
- The English needs to be extensively improved. The authors often use unnecessarily short phrases, and change the tenses (past, present and future) and pronouns (1stand 3rd person).
Thank you for your feedback. We have revised the English in the article, improving it, including the correction of the tenses and pronouns.
- The manuscript lacks images of relevant results on drug delivery systems for lung cancer.
Thank you for your comment. As we understand the importance of images in the review articles, we have included in our manuscript a comprehensive table that highlights the most important aspects of the drug delivery systems, which complements the already existing microscopy images included in the manuscript. These additions aim to provide a thorough representation of the drug delivery systems relevant to lung cancer treatment.
- The authors should include tables that summarize the current drug delivery systems for lung cancer therapy, including the composition, therapeutic agents, functionalization, advantages and disadvantages.
Thank you for your comment. We have included this table in our manuscript.
- Several sections of the paper mention findings and clinical trials without properly citing the respective article. For example, see the section from line 192 to 228.
Thank you for your comment. We have revised the citations in the manuscript
- The authors should reconsider the naming of chapter 4, as it includes a mixture of information related with nanosystems, targeting strategies and drugs that would be better included in other chapters. For instance, all the discussion on targeting should be in chapter 5, while the discussion on commonly used drugs would be more adequate for a different chapter.
Thank you for your insightful comment. In response, we have carefully rearranged Chapters 4, 5, and 6 to enhance the manuscript’s clarity and structure. The sections previously focused on delivery nanosystems have been reorganized under a new Chapter 5, titled Nanocarrier-Mediated Drug Delivery Systems, allowing for a more logical flow of information. Additionally, the chapter formerly entitled Receptor-Based Targetinghas been moved and is now presented as Chapter 4. This reorganization ensures that key concepts are easier to follow and more coherently presented.
- Lines 283 to 286 seem to belong to the introduction.
As the information from 283-286 appears in the introduction section, we have completely eliminated the previously-mentioned lines.
- Despite the authors highlighting the VEGF and VEGFR with an extensive description of their overexpression, the manuscript misses a discussion on nanosystems that explore the targeting of VEGF/VEGFR.
Thank you for your constructive feedback. In response to your suggestion, we have included discussions about nanosystems that specifically target VEGF/VEGFR. This new addition elaborates on the various nanotechnology-based approaches that have been developed to inhibit VEGF/VEGFR signaling, highlighting their potential therapeutic benefits in lung cancer treatment. This ensures that the manuscript now provides a more comprehensive overview of nanosystems targeting these pathways, aligning with the extensive description of VEGF and VEGFR overexpression.
- Considering the topic of the review, a larger discussion on the nanosystems would be expected. However, in each nanosystem subchapter, only few studies are mentioned, not critical discussed, and mostly lack any relation with the topic. Besides, unclear phrases and lack of citations are commonly found as in the case of lines 439 to 448.
Thank you for your feedback. We understand the importance of providing a more in-depth discussion on nanosystems. In response, we have expanded each nanosystem subchapter to include a broader range of studies, along with a critical discussion of their relevance to lung cancer treatment. We have also revised unclear phrases and ensured proper citation throughout
Comments on the Quality of English Language
The authors often use unnecessarily short phrases and change the tenses (past, present and future) and pronouns (1st and 3rd person). Besides, the manuscript displays sections with difficult-to-read phrases.
Thank you for your feedback. We have revised the English in the article, improving it, including the correction of the tenses and pronouns.
Round 2
Reviewer 1 Report
Comments and Suggestions for Authors
This revised manuscript has been improved its quality significantly.
All of my concerns were cleared.
Author Response
This revised manuscript has been improved its quality significantly.
All of my concerns were cleared.
Thank you very much for your positive feedback. We truly appreciate your thoughtful comments and suggestions, which have contributed significantly to enhancing the quality of the work.
Reviewer 2 Report
Comments and Suggestions for Authors
The authors considered the reviewer's comments, and the quality of the manuscript was improved. However, some minor points are suggested:
1) Table 1 should be mentioned in the manuscript text.
2) The manuscript should include images of relevant results in lung cancer of each nanosystem.
Author Response
Dear Reviewer,
Thank you for your valuable comments and suggestions. We have carefully considered your feedback and made the following revisions to the manuscript:
The authors considered the reviewer's comments, and the quality of the manuscript was improved. However, some minor points are suggested:
1) Table 1 should be mentioned in the manuscript text.
Table 1 has been properly cited in the text to ensure clarity and relevance in the discussion of nanocarrier-mediated drug delivery systems.
2) The manuscript should include images of relevant results in lung cancer of each nanosystem.
We have included images representing the relevant results of each nanosystem in the context of lung cancer treatment (now included as Fig 5), which provides a visual summary of the key nanocarrier systems discussed in the manuscript.
We believe these changes have further enhanced the quality of the manuscript, and we look forward to your feedback.